# Urgent Need for Adolescent Physical Activity Policies and Promotion: Lessons from “Jeeluna”

**DOI:** 10.3390/ijerph17124464

**Published:** 2020-06-21

**Authors:** Omar J. Baqal, Hassan Saleheen, Fadia S. AlBuhairan

**Affiliations:** 1College of Medicine, Alfaisal University, Riyadh 11533, Saudi Arabia; ojaved@alfaisal.edu; 2National Family Safety Program, King Abdulaziz Medical City-Ministry of National Guard Health Affairs, Riyadh 11426, Saudi Arabia; hassan_nazmus@hotmail.com; 3Department of Pediatrics & Adolescent Medicine, Aldara Hospital and Medical Center, Riyadh 12714, Saudi Arabia

**Keywords:** adolescent, adolescent health, physical activity, exercise, health surveillance, Kingdom of Saudi Arabia

## Abstract

Physical inactivity is a growing concern in Kingdom of Saudi Arabia (KSA) and globally. Data on physical activity (PA) trends, barriers, and facilitators among adolescents in KSA are scarce. This study aims to identify PA trends amongst adolescents in KSA and associated health and lifestyle behaviors. Data from “Jeeluna”, a national study in KSA involving around 12,500 adolescents, were utilized. School students were invited to participate, and a multistage sampling procedure was used. Data collection included a self-administered questionnaire, anthropometric measurements, and blood sampling. Adolescents who performed PA for at least one day per week for >30 min each day were considered to “engage in PA”. Mean age of the participants was 15.8 ± 0.8 years, and 51.3% were male. Forty-four percent did not engage in PA regularly. Only 35% engaged in PA at school, while 40% were not offered PA at school. Significantly more 10–14-year old than 15–19-year-old adolescents and more males than females engaged in PA (<0.01). Mental health was better in adolescents who engaged in PA (<0.01). Adolescents who engaged in PA were more likely to eat healthy food and less likely to live a sedentary lifestyle (<0.01). It is imperative that socio-cultural and demographic factors be taken into consideration during program and policy development. This study highlights the urgent need for promoting PA among adolescents in KSA and addressing perceived barriers, while offering a treasure of information to policy and decision makers.

## 1. Introduction

Physical activity (PA) when performed regularly, promotes well-being and helps in prevention of various health problems. In children and adolescents, PA plays a particularly critical role in growth and development, while promoting their transition into healthier adults, as lifestyle habits and PA patterns are largely established during childhood and adolescence [1]. It can help children and adolescents “see better, feel better and function better” [2]. Essentially, PA achieves this by helping them build strong bones, maintain healthy weight, improve cardiorespiratory fitness, alleviate symptoms of depression and anxiety, and also reduce the risk of developing several illnesses, such as cancer, type 2 diabetes, heart disease and obesity [2]. Studies have shown that physically active students have better grades, school attendance, cognitive performance and classroom behaviors [3,4]. It is recommended that children and adolescents between the ages of 6 and 17 years have at least 60 min of PA each day [5,6].

Despite widely recognized benefits and establishment of guidelines for PA, adolescents’ PA behaviors have been far from encouraging. Globally, a significant increase in physical inactivity has been observed [7], which has led to increasing incidence of several health problems, imposing a considerable burden upon healthcare systems of many countries. In the United States (US), it was found that only 21.6% of children and adolescents aged between 6 and 19 years attained 60 or more minutes of moderate-to-vigorous PA on at least 5 days per week [8]. In England, only 21% and 16% of boys and girls, respectively were found to meet minimum PA recommendations [9]. Even though the magnitude of the health issues associated with physical inactivity is well-known, relatively few countries conduct physical inactivity surveillance and monitoring [10].

In the Kingdom of Saudi Arabia (KSA), a consequence of rapid socio-economic growth has been striking lifestyle changes. This has had adverse effects on population health, with an increase in physical inactivity and obesity being particularly observed [11]. Prevalence of physical inactivity in the Saudi population has been found to range between 46% and an astonishing 99% in certain segments of the population [11]. A study involving Saudi school students noted high inactivity levels, with roughly only 44% of males and 20% of females being sufficiently active [12]. Consistent with global PA trends [13], males have been reported to participate more frequently in high-intensity PA than females [10].

In parallel with increasing physical inactivity, KSA has also experienced an increasing prevalence of obesity and activities that promote sedentary lifestyle, with a study noting a negative relationship between PA and obesity [14]. The same study noted a significant inverse relationship between physical activity and computer-usage (a sedentary activity), indicating that the more active the individual, the less time spent on computer-usage, with a higher body mass index (BMI) observed amongst those who reported greater computer-usage time [14]. Geographical factors have been shown to influence PA habits, with a study reporting that youths living in rural desert areas were less physically active than those living in urban or rural farm environments [15].

With the dramatic lifestyle transformation, a change in dietary habits and food consumption patterns has also been noted. Calorie-rich foods such as fast food, soft drinks, and processed snacks have become extremely affordable and accessible to children and adolescents [16]. Such food consumption patterns potentially synergize the negative effect of physical inactivity and sedentary lifestyles on population health, in turn contributing to the growing prevalence of obesity, heart disease and other non-communicable diseases. KSA’s hot sun-drenched climate limits physical activity as people prefer staying indoors to avoid exposure to sunlight and heat, though there have been increasing opportunities for indoor places and activities. The widespread vitamin-D deficiency among Saudi children and adolescents has been shown to be influenced by PA and sun exposure [17].

Data on PA trends, determinants, barriers and facilitators among adolescents in Saudi Arabia have been scarce. “Jeeluna” (Arabic for “Our Generation”), a national school-based cross-sectional study involving around 12,500 adolescents in all 13 regions of KSA, found that only 13.7% of adolescents aged 10–19 years engaged in at least 30 min of PA daily [18]. To the best of our knowledge, no other studies on the same have been published exclusively on the adolescent age group (10–19 years), with most published studies focusing solely on older adolescents. This is of limited benefit when attempting to extrapolate research findings to the development of policies, programs and services promoting PA focused particularly on the adolescent age group.

The aim of this study is to identify the sociodemographic characteristics of adolescents who participate in PA in KSA, as well as their associated health and lifestyle behaviors by gender. Such information will support program and policy development that address adolescents’ local needs in an evidence-based manner.

## 2. Materials and Methods

### 2.1. Study Design and Participants

For this current analysis, we utilized data from the Jeeluna study database. Jeeluna was a cross-sectional, school-based, nationally representative study conducted in Saudi Arabia. Participants included intermediate and secondary school students, aged 10–19 years from all 13 regions of the country.

### 2.2. Procedures

A stratified, cluster random sampling procedure was used. The detailed sampling methodology has been previously published [18]. Any male/female, intermediate/secondary, public/private school in a Saudi Arabian city/town that functions during the day was eligible to participate in the study. Exclusion criteria included evening schools and schools that served students with special needs. Data collection involved: (1) administration of self-administered questionnaire; (2) anthropometric measurements; and (3) blood sampling for laboratory investigations. Data collection teams operating in all regions received standardized training.

The study received approval from the Institutional Review Board at the King Abdullah International Medical Research Center (KAIMRC) (RC08-092), as well as the Ministry of Education (MOE). It was necessary to seek permission from participating schools’ principals, active written parental consent and student assent; participants were given the option to opt out of blood sampling.

For the purpose of this secondary analysis, the focus was on the PA section. It was assessed by asking students how many days in the past week they had engaged in exercise for at least 30 min each day. The cutoff of 30 min was used instead of the recommended 60 min, as PA trends have shown that most children and adolescents do not meet the recommended minimum, thus supporting the use of a lower cutoff in the analysis.

### 2.3. Measurements

The questionnaire was guided by the Youth Risk Behavior Survey [19] and the Global School-based Student Health Survey [20]. The global definition of health was used, so multiple domains were included and addressed in the questionnaire, including: (1) family; (2) education/schooling; (3) nutrition/dietary behaviors; (4) physical activity; (5) safety; (6) sleep; (7) violence and bullying; (8) tobacco and substance use (including alcohol use); (9) health; (10) health services; and (11) health knowledge. Students were given assurance of the anonymity and confidentiality of their responses.

Adolescents who engaged in PA for 1 or more days in a week were considered to “engage in PA”. The section also included questions on sedentary activities (TV, internet, video games and cell phone use). Responses indicating time spent on each activity to be more than 2 h daily were considered to be part of the “sedentary lifestyle” category. With regard to main meals (breakfast, lunch and dinner), a value of “3” meals a day was considered healthy. Everything else, whether lower (none, 1 or 2 meals a day) or higher (4 or >4 meals a day), was put in the “unhealthy” category. We considered guidelines from the American Academy of Pediatrics (AAP) that refer to 3 meals a day as the target for healthy nutrition [21]. The 5 response choices pertaining to absence from school were grouped into 3 categories: “Not absent”, “Rarely absent”, and the third category including “sometimes”, “often” and “always.”

The “Mental health” variable was created based on responses to the questionnaire items asking about the frequency of symptoms reflecting depression or anxiety over the past 12 months. “Never”, “rarely” and “sometimes” were considered as “No”, and “most of the times” and “always” were considered as “Yes.” The anthropometric measurement from the study database included in this analysis was BMI, which was based on measured weights and heights. BMI was classified as “Underweight”, “Normal”, “Overweight” and “Obese”, based on Center for Disease Control and Prevention BMI charts for gender and age [22,23].

### 2.4. Data Analysis

The first step in the analysis was the descriptive analysis. Participants were described in terms of their selected socio-demographic status. Means along with standard deviations and percentages were calculated for continuous and categorical variables, respectively. The relationship between engaging in PA and school performance, health status, and health risk behaviors was calculated using chi-square test. A significance level of less than 0.05 was used for all statistical tests. All data were analyzed using SPSS version 25.0.3 (IBM Corp, Armonk, NY, USA).

## 3. Results

Participant characteristics are listed in Table 1. Mean age of the participants was 15.8 ± 1.8 years, and 51.3% were male. Most of the participants (97.9%) lived in urban areas. Little over half (53.1%) of the participants fell within the “normal” BMI range; similarly, just over half reported engaging in PA (53.4%). Only about one-third (35.4%) engaged in PA at school.

Nationally, 92.7% of males and 7.3% of females were offered PA in schools. Out of the 13 regions, Jizan had the highest mean number of PA days per week in total (2.4 ± 2.8) as well as among males (3.3 ± 2.8). Jizan also boasts the highest proportion of total (44.7%) and male students (99.6%) engaging in PA in school. Interestingly, a stark contrast is observed in PA trends among females in Jizan, which has the lowest mean number of PA days per week in females (0.8 ± 1.7) among all regions, as well as the lowest proportion of females engaging in PA in school (0.4%). Female PA trends were dramatically better in Aljouf, which had the highest proportion of females engaging in PA in school (53.2%), and second highest mean number of PA days (1.8 ± 2.4). Detailed region-wise PA trends are displayed in Table 2.

Figure 1 reflects a decline in PA relative to age, with mean PA days per week of 2.7 at 12 years among males dropping to 2.1 by age 18, before rising to 2.4 by 19 years of age. In females, PA steadily declined between 12 and 19 years of age from 1.6 to 1. Significant differences in age, gender, and areas of residence, were found in terms of engagement in PA, with younger adolescents (59.6% vs. 53.5%, *p* < 0.01), males (68.3% vs. 48.7%, *p* < 0.01), and adolescents living in urban areas (55.1% vs. 47.1%, *p* < 0.01) being more likely to engage in PA. PA habits were not significantly different between Saudi and non-Saudi adolescents, as well as between the four BMI groups, with proportions of adolescents engaging in PA ranging between 54% and 57% across the BMI groups (Table 3).

Table 4 represents the relationship between engaging in PA and school performance, health status, and health risk behaviors. Adolescents who engaged in PA were more likely to eat healthy food (99.1% vs. 98.3%, *p* < 0.01), less likely to live a sedentary lifestyle (63.9% vs. 67.3%, *p* < 0.01), and less likely to be absent from school (26.6% vs. 20.3%, *p* < 0.01). With regards to mental health, symptoms reflective of depression (12.2% vs. 17%, *p* < 0.01) and anxiety (5.9% vs. 7.8%, *p* < 0.01) were found to be less prevalent among adolescents who engaged in PA. Prevalence of chronic diseases as well as tobacco/substance use was not significantly different between the two groups. No significant difference was found in number of students with “excellent” academic performance among those who engage in PA and those who do not.

## 4. Discussion

Research activity on PA and lifestyle habits has not matched the rapid pace at which lifestyle changes have been occurring in the KSA. This study, based on the Jeeluna national school-based study, sheds light on the status quo of PA among the adolescents of KSA and related health and lifestyle habits. Over 40% of adolescents in our study did not engage in PA at all. The remaining adolescents were part of a wide-ranging category that included those who engaged in PA once a week as well as those who engaged in PA daily. Twenty percent of males and a worrying 59% of females did not engage in PA regularly. Locally reported studies that utilized objective PA measurement techniques showed that 60% of Saudi children and 71% of youths did not engage in PA of sufficient frequency and duration [24,25].

To put local estimates including ours in perspective, it is suitable to compare them with PA estimates from other countries. A systematic assessment of literature on PA in Arab countries revealed that physical inactivity was alarmingly high in children and adolescents, reaching about 80% in all national surveys except Tunisia [26]. A Finnish study reported that 59% of boys and 50% of girls aged 15–16 years reported engaging in 60 min or more of PA per day; these estimates dropped to 23% (boys) and 10% (girls) when moderate-to-high physical activity was considered [27]. In the US, the Youth Risk Behavior Surveillance revealed that only 18.4% of adolescents met the PA guidelines [28]. The Global School-based Student Health Survey (GSHS), which included over 72,000 young people across 34 countries, showed that only 24% of boys and 15% of girls engaged in sufficient PA to meet recommendations [29]. Recently, WHO published a pooled analysis of survey data reported by 1.6 million 11 to 17-year-old adolescents across 146 countries. It found that more than 80% of school-going adolescents globally did not meet current daily PA recommendations, including 85% of girls and 78% of boys [30].

These worrying estimates of inactivity are of a great concern as inactivity has been associated with increased risk of development of cardiovascular and metabolic health conditions [2]. Modern life has paralleled technological advances, reflected in increased screen time for everyone, whether for work or leisure. This lifestyle has aimed to minimize efforts and systematically reduce energy expenditure and, in the process, to discourage physical exertion. We found that 67% of adolescents who did not exercise pursued a sedentary lifestyle. Spending time on the previously mentioned sedentary activities soaks up valuable time that could otherwise be utilized in health-promoting activities such as regular exercise. Amusingly, a lack of time has been regularly reported by individuals for their lack of PA [31,32]. Other factors that contribute to the reported adolescent inactivity rates in KSA may include lack of PA programs in schools, hot weather conditions, over-dependence on cars rather than walking even for short distances, the surrounding built environment which is not pedestrian friendly, poor peer and family support and socio-cultural barriers, which particularly impact females.

We found that males were significantly more likely to engage in PA than females. This finding is consistent with findings of several other studies [16,33,34]. This could be attributed to cultural beliefs and lifestyle patterns in KSA that possibly discourage participation of females in recreational activities outside the house. It is perhaps due to these socio-cultural differences that experts have suggested a specific gender-based consideration when making PA promotion recommendations [35]. Although males have been generally considered to have better opportunities for recreational activities including exercise than females, rapid changes have fortunately been witnessed in the past 1–2 years with more sports opportunities for females being made available in the country. The opening of female-only gyms, sports clubs, and female national sports teams have encouraged more women to engage in regular exercise. Interestingly, our study found that females who exercise were significantly more likely to eat heathy food and less likely to live a sedentary lifestyle than females who did not exercise, while there was no such significant difference between the two male groups regarding healthy eating and sedentary lifestyle. This could reflect active consciousness and awareness regarding healthy eating and lifestyle habits among females who exercise regularly. On the other hand, males seem to manage to get sufficient PA, possibly through school and recreational activities while still personally not living an active healthy lifestyle.

Our study found a stark contrast between PA offered to males and females in schools. For example, in Jizan, 99.6% of male adolescents reported being offered PA at school, while only a shocking 0.4% of females reported the same. A local study showed that males from public schools were more active than males at private schools, whereas the opposite was true for females; they were more active in private schools [12]. While PA programs in female public schools are nearly nonexistent, private schools provide students with opportunities for regular exercise, encouraging an active and healthier lifestyle. Our findings are in agreement with this observation as most of the schools in Jizan belong to the public education system. We also reported Jizan to have the highest mean number of PA days per week among males in the whole country, while the lowest among females. Jizan is considered to be relatively more conservative when compared with the bigger cities of Jeddah and Riyadh, and thus, the cultural and lifestyle barriers to female participation in PA mentioned earlier may be augmented in this region.

We also found that adolescents aged 10–14 years were more likely to engage in PA than older adolescents aged 15–19 years. PA decline in adolescence has been previously observed [36], a trend that seems to continue into adulthood [37,38]. Increasing academic workload in secondary school could contribute to this decline, and factors such as more time spent on sedentary activities [39] as well as taking up part-time jobs [40] could also be related. This highlights the need for qualitative studies to clearly understand the reasons for this decline and to promote PA and other healthy habits in adolescence, a stage of building habits, many of which last a lifetime.

There was no significant difference in PA habits between Saudi and non-Saudi adolescents, as well as between different BMI groups. These interesting findings counter expectations, as it would generally be convenient to believe that expats from more open societies with lesser socio-cultural barriers would be living a more active lifestyle. This highlights the impact of lifestyle changes an individual adopts when living in a foreign country.

Among all of the benefits of PA and its positive effects on general health, its impact on mental health is well recognized. Exercise is known to boost self-esteem, body image and overall mood, in line with our findings of adolescents who engage in PA feeling sad or hopeless significantly less frequently than those who do not engage in PA [41]. Exercise is widely recommended by health authorities for the prevention and treatment of various non-communicable diseases, although the above mentioned benefits may depend on type, timing, and intensity of PA. Physical activity has been well-studied in the area of depression, with a study reporting 20–33% lower odds of depression in the active groups in prospective cohort studies [42]. For tasks involving more complex executive functioning, exercise was found to be associated with enhanced cognitive functioning in a meta-analysis of randomized controlled trials looking into exercise training studies in adults aged 55–80 [43,44].

We found that students who engaged in PA were significantly more likely to not be absent from school. Based on 2005–2008 NHANES (National Health and Nutrition Examination Survey) data, it was found that excessive TV watching and inactivity as well as high activity levels (>7 times per week) were independently associated with severe school absenteeism [45]. As discussed previously, excessive screen time could be taking away valuable time from adolescents that they could have spent being more physically active. Moreover, excessive TV watching can have various negative effects on a child’s cognitive and socio-emotional development. Having discussed the benefits of PA on mental health, low PA levels could hamper an adolescent’s participation in school, such that school is something they do not look forward to due to negative self-esteem, body image, feelings of sadness and worry, or lack of motivation.

Consequently, it is indeed worrying that despite the myriad of known health benefits, PA trends among our adolescents remain poor. PA, even in small amounts can considerably enhance physical and psychological well-being. School-based PA programs can address the enormous discrepancy between female and male PA habits in KSA by improving availability and accessibility to PA opportunities. Health care professionals, starting from the primary health care system and school health system, must convey the benefits of regular physical activity to individuals and their families and do everything in their capacity to facilitate their participation in PA. The American Heart Association emphasizes that “the advice from healthcare professionals significantly influences adoption of healthy lifestyle behaviors, including regular PA, and can increase satisfaction with medical care” [46]. Several medical institutions, particularly in the US have begun assessing PA as a vital sign tool (indicator of general physical condition) to identity individuals who could benefit from prompt counseling or appropriate referral. Parents and guardians play a critical role in ensuring adolescents at least meet the minimum recommended amount of PA as per latest guidelines. The remarkable technological literacy of this young generation can be positively utilized by promoting active e-sports video games (e.g., Wii Sports) and health and fitness tracking applications. Above all, national health authorities, particularly the Ministry of Health, Ministry of Education, and Sports Authority must strategize an efficient multi-level action plan by taking into consideration existing data on local PA trends and lifestyle habits among adolescents, including results of this landmark study. This will also significantly contribute towards lowering the currently estimated total costs attributable to physical inactivity in KSA.

The Jeeluna study included over 12,000 adolescents, making it the largest study on adolescent health in the region to date. Additionally, the study was conducted in all 13 regions of KSA; the national representativeness of the study population enhances its generalizability. However, the study may be viewed in light of a few limitations. The study questionnaire was self-administered and may be associated with recall bias. Physical activity was not measured objectively, nor was the type of PA reported taken into consideration. In search of an explanation for the distortion of the graph line at 11 years of age, we found that the study population aged 11 years was small in number. The questionnaire did not take into consideration food portions when asking about frequency of meals. Nonetheless, the questionnaire was put through several rounds of expert review and was pilot tested for clarity and comprehension amongst the target respondent group. The study thus sets a benchmark for future adolescent health research in the country and the wider region.

## 5. Conclusions

The Jeeluna study highlights the urgent need for promotion of PA among adolescents in KSA and addressing perceived barriers. It is imperative that socio-cultural and demographic factors be taken into consideration during program and policy development. Schools play a critical role in improving PA habits among adolescents and remain an important focus of action, particularly for female PA promotion. Lack of PA, unhealthy food and sedentary activities are not isolated, independent health risk factors and are often found clustered among adolescents, synergistically increasing risk of disease and poor health several-fold. In order to be effective, these issues must be addressed in a cross-sectoral approach, involving multiple ministries and other stakeholders.

The study sheds light on the urgent and important issue of physical inactivity, which should be of utmost national and international interest. Our study offers valuable information to policymakers, educators and health professionals involved in the progression of adolescent health to understand young people’s health in their social context. Further studies are required to be able to assess trends in PA and to evaluate impact of more recently implemented programs in the country and hence support strategies for further promotion of PA and the reduction of sedentary behavior among adolescents.

## Figures and Tables

**Figure 1 ijerph-17-04464-f001:**
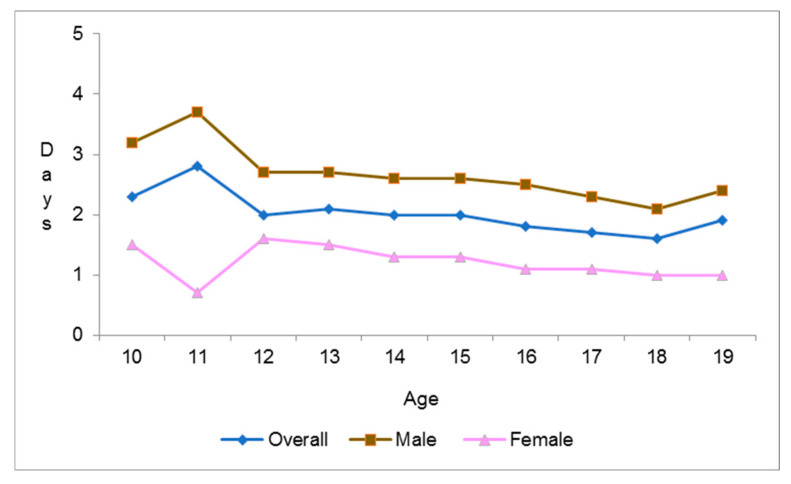
Mean no. of days/week spent on >30 min of physical activity relative to age.

**Table 1 ijerph-17-04464-t001:** Characteristics of the participants (*n* = 12,463).

Characteristic	Number (%) ^¥^
**Age (mean)**	
Mean ± SD	15.8 ± 1.8
Gender	
Male	6398 (51.3)
Female	6065 (48.7)
**Areas of residence**	
Urban	12,205 (97.9)
Rural	258 (2.1)
**Nationality**	
Saudi	10,318 (82.8)
Non-Saudi	1493 (12.0)
**Grade**	
Intermediate	6142 (49.3)
Secondary	6321 (50.7)
**BMI result**	
<5th centile	1858 (14.9)
5–<85th centile	6616 (53.1)
≥85th–<95th centile	1712 (13.7)
≥95th centile	1920 (15.4)
**Engage in physical activity**	
Yes	6653 (53.4)
No	5450 (43.7)
**Engage in physical activity (sports) at school**	
Yes	4412 (35.4)
No	2624 (21.1)
No physical activity at school	5052 (40.5)

^¥^ Percentages may not add up to 100 due to missing data.

**Table 2 ijerph-17-04464-t002:** Physical activity relative to region.

Region	Mean No. of Days/Week Spent on >30 min of Exercise			Regional Sample	Physical Activity (Sports) in School Number (%)		
	Total	Male	Female		Total PA	Male	Female
**Riyadh**	1.7 ± 2.3	2.2 ± 2.4	1.3 ± 2.2	2749	876 (31.8)	853 (97.4)	23 (2.6)
**Qasim**	1.5 ± 2.2	2.0 ± 2.4	1.0 ± 1.8	734	243 (33.1)	239 (98.4)	4 (1.6)
**Makkah**	2.0 ± 2.5	2.7 ± 2.7	1.0 ± 1.9	2906	1099 (37.8)	1074 (97.7)	25 (2.3)
**Madinah**	1.8 ± 2.4	2.5 ± 2.6	1.0 ± 1.9	943	318 (33.7)	312 (98.1)	6 (1.9)
**Eastern province**	1.8 ± 2.4	2.4 ± 2.6	1.2 ± 2.0	1851	683 (36.8)	577 (84.5)	106 (15.5)
**Tabuk**	1.7 ± 2.3	2.2 ± 2.5	1.3 ± 2.0	342	111 (32.4)	81 (73.0)	30 (27.0)
**Aljouf**	1.6 ± 2.2	1.2 ± 2.0	1.8 ± 2.4	361	111 (30.7)	52 (46.8)	59 (53.2)
**Hail**	1.8 ± 2.3	1.7 ± 2.3	2.0 ± 2.4	358	124 (34.6)	76 (61.3)	48 (38.7)
**Northen borders**	1.8 ± 2.4	2.7 ± 2.7	1.2 ± 1.9	222	73 (32.8)	69 (94.5)	4 (5.5)
**Albaha**	1.5 ± 2.3	1.8 ± 2.5	1.3 ± 2.2	314	50 (15.9)	43 (86.0)	7 (14.0)
**Aseer**	2.2 ± 2.6	3.0 ± 2.8	1.3 ± 2.1	773	324 (41.9)	316 (97.5)	8 (2.5)
**Jizan**	2.4 ± 2.8	3.3 ± 2.8	0.8 ± 1.7	606	271 (44.7)	270 (99.6)	1 (0.4)
**Najran**	2.0 ± 2.4	2.3 ± 2.4	1.6 ± 2.3	304	129 (42.4)	127 (98.4)	2 (1.6)
**Overall**	1.9 ± 2.4	2.4 ± 2.6	1.2 ± 2.1	12,463	4412 (35.4)	4089 (92.7)	323 (7.3)

**Table 3 ijerph-17-04464-t003:** Relationship between engaging in physical activity (PA) and demographic characteristics (*n* = 12,463).

Engagement in PA	Age			Gender			Areas of Residence			Nationality			BMI				
	10–14	15–19	*p* Value	Male	Female	*p* Value	Urban	Rural	*p* Value	Saudi	Non-Saudi	*p* Value	Under-Weight	Normal	Over-Weight	Obese	*p* Value
Engage in PA																	
Yes	1905 (59.6)	4673 (53.5)	<0.01	4266 (68.3)	2387 (40.7)	<0.01	6532 (55.1)	121 (47.1)	<0.05	5575 (54.9)	835 (56.6)	NS	1010 (55.0)	3541 (54.4)	962 (56.9)	1057 (56.0)	NS
No	1293 (40.4)	4061 (46.5)		1976 (31.7)	3474 (59.3)		5314 (44.9)	136 (52.9)		4579 (45.1)	640 (43.4)		826 (45.0)	2964 (45.6)	729 (43.1)	830 (44.0)	

**Table 4 ijerph-17-04464-t004:** Relationship between engaging in PA and school performance, health status, and health risk behaviors among adolescents in Saudi Arabia (*n* = 12,463).

Variable	Total			Male			Female		
	PA	No PA	*p*-Value	PA	No PA	*p*-Value	PA	No PA	*p*-Value
School performance									
Excellent academic performance	2845 (43.3)	2383 (44.2)	NS	1709 (40.5)	766 (39.1)	NS	1136 (48.2)	1617 (47.0)	NS
Not absent from school	1745 (26.6)	1095 (20.3)	<0.01	1282 (30.4)	533 (27.3)	<0.05	463 (19.7)	562 (16.3)	<0.01
Chronic diseases	566 (8.8)	438 (8.2)	NS	386 (9.4)	174 (9.1)	NS	180 (7.7)	264 (7.7)	NS
Mental Health									
Sadness/frustration	800 (12.2)	915 (17.0)	<0.01	388 (9.2)	242 (12.4)	<0.01	412 (17.5)	673 (19.6)	<0.05
Anxiety	383 (5.9)	416 (7.8)	<0.01	171 (4.1)	117 (6.0)	<0.01	212 (9.1)	299 (8.8)	NS
Healthy food	6584 (99.1)	5344 (98.3)	<0.01	4231 (99.3)	1958 (99.2)	NS	2353 (98.7)	3386 (97.7)	<0.05
Sedentary lifestyle	4222 (63.9)	3643 (67.3)	<0.01	2653 (62.7)	1246 (63.5)	NS	1569 (66.1)	2397 (69.4)	<0.01
Tobacco/substance use	1658 (25.6)	1350 (25.3)	NS	1041 (25.1)	521 (27.0)	NS	617 (26.4)	829 (24.3)	NS

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
