# Peer review of "Urgent Need for Adolescent Physical Activity Policies and Promotion: Lessons from “Jeeluna”"

_ijerph, 2020, doi:10.3390/ijerph17124464_

Round 1

Reviewer 1 Report

Abstract

Line 26 – please consider changing “highlights the urgent need for promotion of PA among adolescents” to “highlights the urgent need of promoting PA among adolescents”

Introduction

This section seems well written and easy to follow. Some very small things were noticed:

Line 47 – The authors could abbreviate the expression “physical activity”

Line 63 – It was the first time that BMI appears, please right “body mass index” to clarify

Line 79 – Here was stated “over 12.500” and in the abstract and discussion “over 12.000”

Material and Methods

2.1. Design and participants

Please provide information about the sex of the sample and the age distribution of the overall sample.

Line 103 to 106 – please consider changing this paragraph to the “study design” session

Line 107 – Please consider changing is” to “was”

Line 123 – why you use 3 meals as “healthy”?

Line 131 – now BMI could be abbreviated, as it was included the all meaning in the introduction session

Results

Line 142-143 – in my opinion, the information in the second sentence of this lines should be included in the methods session

Discussion

Line 190, 200 and 286 – The authors could abbreviate the expression “physical activity”

Line 266-267 – please provide a reference

Line 272-275 – instead of including studies with adults, the authors could include studies with youth (the same topic)

The sample include a large category of ages, therefore, as a suggestion, it would be better to divide at least in two groups. As the authors indicated in the discussion session, the patterns between 10 and 14 and between 15 and 19 years of age were different. Moreover, I would also suggest to include a third group, a group that engage and perform the minimum physical exercise recommended by ACSM, thus, the sample will be divided in 1) those that did not perform any exercise; 2) those that are engage with exercise (at least 1 day for 30min); 3) those that perform the recommended guidelines.

Reviewer 2 Report

The present is characterized by an ever-decreasing volume of realized physical activities in children and adolescents. This is mainly due to the offer and conditions for the implementation of the PA and, of course, the cultural tradition of the region. Therefore, any study that tries to describe the current situation is current and necessary. The region in which the study originated and its scope need to be highly evaluated. On the other hand, I have some key comments on the study. 1. why the PA volume of 30 minutes was chosen, which are values more valid for adults, for children and adolescents, one exercise unit is more likely to be required for a duration of 60 minutes. 2. what were the physical activities and conditions for implementation, which is necessary for the interpretation of the results. 3. PA was implemented within the framework of the school PE. 4. how body mass and height measurement was carried out, which is essential for the calculation of BMI. 5. For the determination of the weight status were used charts of the American CDC, are similar graphs available in Saudi Arabia?. The use of inadequate development charts can significantly influent the results mainly in the area of overweight and obesity transition.

I don't formally understand the sentence on lines 121-122. In conclusion, I recommend more concrete and suggest how to influence the lifestyle of adolescents, so that there is a change of their sedentary to an active lifestyle.

Round 2

Reviewer 1 Report

Dear Authors,

the role of a reviewer is to try to help to improve the authors’ work, not just criticize for criticizing. So, when you don't agree, which is entirely legitimate, please justify it, but with the literature and not just with "I think so", or just ignore.

Point 6: Line 103 to 106 – please consider changing this paragraph to the “study design” session – For this comment, it doesn’t make any sense to include an “Ethical Considerations” session alone. I suggested including in the design session to confirm that it was approved.

Point 8: Line 123 – why you use 3 meals as “healthy”? – I just ask for a valid reference to justify your option, as it is not always considered the ideal and healthiest, especially in adolescents in development.

Point 14: The sample include a large category of ages, therefore, as a suggestion, it would be better to divide at least in two groups. As the authors indicated in the discussion session, the patterns between 10 and 14 and between 15 and 19 years of age were different. Moreover, I would also suggest to include a third group, a group that engage and perform the minimum physical exercise recommended by ACSM, thus, the sample will be divided in 1) those that did not perform any exercise; 2) those that are engage with exercise (at least 1 day for 30min); 3) those that perform the recommended guidelines. – I still think that the Authors should split the sample, as maturation could make a huge difference in the results. The Author’s answer suggests that we should adjust the statistical analysis to the data and not search for an answer to a problem, which in my opinion, will skew the conclusion of the study.

Author Response

Response to Reviewer 1 Comments

Point 6: Line 103 to 106 – please consider changing this paragraph to the “study design” session – For this comment, it doesn’t make any sense to include an “Ethical Considerations” session alone. I suggested including in the design session to confirm that it was approved.

Response 6: Thank you for your comment. Kindly note that the Ethical Considerations section is a standard section in the Methodology (https://www.intechopen.com/online-first/research-design-and-methodology). This is where IRB approvals and other related items are included. We therefore do not find it appropriate to include this component (Lines 103-106) in the Study Design section.

Point 8: Line 123 – why you use 3 meals as “healthy”? – I just ask for a valid reference to justify your option, as it is not always considered the ideal and healthiest, especially in adolescents in development.

Response 8: Thank you for your comment. Indeed, meal frequency is only one of the factors that determine meal healthiness. We considered Pediatric guidelines that refer to 3 meals a day as the target for healthy nutrition (https://brightfutures.aap.org/Bright%20Futures%20Documents/BF4_HealthyNutrition.pdf). Any less than that would indicate skipping out on one of these. For instance, several previous studies have shed light on the impact of eating breakfast on various health indicators, such as health-related quality of life (HRQOL), maintenance of a healthy BMI and risk of obesity. Meal frequency is only a part of the story, with other factors such as portion size and quality of meal completing the picture. We did not have data on the meal portion sizes, and thus explicitly mentioned that as one of our study limitations. 

Point 14: The sample include a large category of ages, therefore, as a suggestion, it would be better to divide at least in two groups. As the authors indicated in the discussion session, the patterns between 10 and 14 and between 15 and 19 years of age were different. Moreover, I would also suggest to include a third group, a group that engage and perform the minimum physical exercise recommended by ACSM, thus, the sample will be divided in 1) those that did not perform any exercise; 2) those that are engage with exercise (at least 1 day for 30min); 3) those that perform the recommended guidelines. – I still think that the Authors should split the sample, as maturation could make a huge difference in the results. The Author’s answer suggests that we should adjust the statistical analysis to the data and not search for an answer to a problem, which in my opinion, will skew the conclusion of the study.

Response 14: Thank you for your comment. The aim of this study is to identify the sociodemographic characteristics of adolescents who participate in PA in KSA, as well as their associated health and lifestyle behaviors by gender. To address the specific aim, we categorized PA into two groups - those who engaged in PA and those who did not. In this paper, we did not focus on level/intensity of PA. We will certainly consider implementing your ideas in future publications.     

Round 3

Reviewer 1 Report

Dear authors,

as you clarified my concerns, I am satisfied with this version.

Best regards,

Ana Filipa Silva